# Climate resilient adaptation of coastal amenities for wellbeing

Charlotte Lyddon [ID]

Department of Geography and Planning, University of Liverpool, UK

Blue spaces; Climate change adaptation; Coastal wellbeing; Environmental governance; Coastal hazards

**Corresponding author:**
Charlotte Lyddon;
Email: c.e.lyddon@liverpool.ac.uk

## Abstract

Blue spaces are recognised as vital public resources that support human health and wellbeing through pathways such as physical activity, psychological restoration and social connection. Coastal environments are especially distinctive for their multisensory stimuli, expansive vistas and cultural significance, with benefits shaped by environmental quality, accessibility and usability. Climate change threatens these benefits via extreme events, such as storm surges and flooding, and gradual processes, including rising sea levels, erosion and warming temperatures, which can also alter people's perceptions, engagement patterns and cultural connections. A conceptual framework is proposed to explore the mechanisms through which climate change affects coastal amenities. This framework offers a structured approach to understanding how environmental processes, interventions and social factors interact to shape health and wellbeing outcomes, and identifies where, why and at what scales adaptation interventions can be most effectively applied. Adaptation measures can help sustain wellbeing benefits under climate hazards by reinforcing positive feedbacks, such as stewardship and investment in amenities, while mitigating negative feedbacks from environmental degradation or overuse. Climate-informed adaptation of coastal amenities, integrating ecological, social and governance considerations, is essential to preserve quality, access, usability and the equitable delivery of health benefits. The framework therefore offers both a theoretical basis for understanding climate–wellbeing interactions and a practical tool to support internationally relevant interventions, policy development and co-designed justice-sensitive adaptation strategies that sustain the health, cultural and social value of blue spaces under climate change.

## Impact statement

This research delivers a novel conceptual framework that clarifies how climate change threatens the wellbeing benefits of coastal blue spaces and provides practical pathways for safeguarding them. Coastal amenities, including beaches, promenades, wetlands and heritage sites are vital public goods that promote health through physical activity, psychological restoration and social connection. Here we focus on broadly relevant climate hazards including sea-level rise, erosion, flooding and heatwaves which undermine these benefits indirectly by degrading environmental quality, accessibility and usability, which are the pathways through which blue spaces support wellbeing. The framework also highlights dynamic feedbacks between coastal amenities and climate change: well-maintained, resilient amenities can encourage stewardship, investment and sustainable use that enhance adaptive capacity, whereas neglected or degraded sites can accelerate environmental decline, erode community engagement and amplify climate vulnerabilities. By making these mechanisms and interactions explicit, the framework equips policymakers, planners and community organisations with a structured tool to anticipate and address climate-driven risks to coastal environments. National and regional policymakers can use this evidence to prioritise investment in climate adaptation for coastal amenities, ensuring that blue spaces remain safe, inclusive and restorative for future generations. This strengthens the case for treating coastal environments as critical public health infrastructure.

## Introduction

Coastal amenities comprise a diverse range of environments and features that provide recreational, cultural, and ecological value to people, including beaches, estuaries, dunes, wetlands and heritage sites. Many of these amenities fall within the broader category of blue spaces; outdoor environments, natural or manmade, that prominently feature water and are accessible to people either directly (in, on or near water) or indirectly (through sight, sound or other sensory experience) (Environment Agency, 2020; White et al., 2020). Blue spaces are typically considered public goods, valued for their multifunctional roles rather than designed for a single purpose (DEFRA, 2023). While often appreciated for their natural qualities, blue spaces are frequently shaped or constrained by grey infrastructure, such as

seawalls, promenades and harbours, which can influence both their functional and aesthetic characteristics (Potter et al., 2023).

Coastal environments are distinctive within the blue space category due to their multisensory stimuli, dynamic land–sea edges, expansive horizons, and deep cultural embedding (Tsai et al., 2023; Woodroffe et al., 2023). These features amplify pathways linking blue space exposure to improved health and wellbeing (White et al., 2016; Britton et al., 2020; Garrett et al., 2023) open vistas enhance stress recovery, dynamic edges encourage exploration and physical activity and culturally meaningful sites can foster social cohesion and a sense of belonging (Gascon et al., 2017; Georgiou et al., 2021). Health and wellbeing benefits can be conceptualised as restorative, the replenishment of capacity for health and wellbeing or instorative, relating to the development of capacities over time (White et al., 2020). Evidence suggests that proximity to coastal blue spaces is associated with measurable improvements in both mental and physical health (Grellier et al., 2017; Jarratt et al., 2022), with more frequent or longer engagement generally conferring greater benefits (White et al., 2021; Bray et al., 2022; Geary et al., 2023).

Beyond wellbeing, coastal amenities carry significant economic value through tourism and leisure (Picken, 2025). In Great Britain alone, coastal tourism attracts over 270 million day visitors annually, supporting 285,000 jobs (Elliott et al., 2018), while the OECD predicts that marine and coastal tourism will generate US$777 billion globally by 2030 (OECD, 2016). Despite this economic significance, many coastal towns remain among the most deprived communities, highlighting persistent social inequalities (Wenham, 2022; Fiorentino et al., 2024). Access to the benefits of blue spaces is uneven, constrained by factors such as inadequate infrastructure, seasonal employment patterns, affordability and physical or cultural accessibility (Geiger et al., 2023). Investing in accessible and inclusive coastal amenities therefore represents an opportunity to enhance wellbeing for both local populations and wider tourist communities.

Understanding the wellbeing benefits of coastal amenities requires separating the characteristics of the environment itself from the opportunities for individuals to engage with it. Amenity supply refers to the intrinsic quality of the resource – such as water and habitat conditions, alongside supporting infrastructure, including promenades, signage and visitor facilities (Boto-García and Leoni, 2023). Amenity access and uptake, in contrast, capture the capacity of individuals to reach and benefit from these spaces, influenced by safety, affordability, physical accessibility and social or cultural factors (Orchard et al., 2025).

These dynamics are increasingly shaped by climate change. Global temperatures have risen by approximately 1.7°C per century since 1970, and there is a greater than 50% chance that warming will reach or exceed 1.5°C by 2040 (Intergovernmental Panel on Climate, 2021). Global sea levels have risen ~21 cm since 1900, with projections under high-emission scenarios suggesting increases of 0.63–1.02 m by 2100 (Oppenheimer et al., 2019). For the United Kingdom, mean sea level is expected to rise between 0.29 and 0.70 m under low emissions and 0.53 and 1.15 m under high emissions, with regional variability (Palmer et al., 2018). Concurrently, extreme weather events, including intensified storms (Walsh et al., 2016; Fowler et al., 2021) longer storm seasons (Tamarin-Brodsky and Kaspi, 2017) and compound events (Zscheischler et al., 2018) are projected to increase in frequency and severity, posing escalating risks to coastal environments and communities.

Many coastlines worldwide face similar pressures including accelerating sea-level rise, more frequency and intense storms, shifting coastal morphology, ageing infrastructure and social inequalities (Nicholls and Cazenave, 2010; Perks et al., 2023). At the same time, coastal management frameworks are increasingly recognising that wellbeing, access and public health must sit alongside traditional priorities such as ecological conservation and flood protection in adaptation planning. As countries refine their adaptation approaches under frameworks such as the EU Strategy on Adaptation to Climate Change and the Sendai Framework, tools that connect climate processes to human wellbeing are becoming essential. Against this backdrop, this review synthesises current understanding of how climate change is reshaping coastal amenities. It examines the interactions between environmental processes, blue spaces and human experience, with particular attention to how climate change influences the availability, quality and equitable access to these uniquely valuable coastal environments.

### Conceptual framework

Conceptualising the relationships between climate change, coastal amenities and human wellbeing provides a structured approach for analysing complex pathways and identifying points for intervention (Figure 1). The impacts of climate change on coastal environments occur across multiple timescales, from short-term, extreme events such as flooding to longer term, gradual processes such as erosion, habitat loss and ecosystem shifts (Dolan and Walker, 2006; Elliott et al., 2014; Neumann et al., 2015). These changes threaten both the quality and availability of coastal amenities. Climate hazards do not just affect wellbeing directly but operate primarily through their impacts on the natural and built environment, which in turn shape environmental attributes and access and usability. These channels determine how health and wellbeing benefits are realised. Even when a blue space exists, its environmental quality dictates whether people can access and use it safely. Without access, no benefits can occur; without usability, restorative benefits may be limited to visual exposure, and instorative benefits are unlikely to develop. While degraded or unmanaged blue spaces may still provide some passive benefits, climate adaptation can enhance the magnitude, reliability and sustainability of health benefits by maintaining safe and meaningful engagement (Bell et al., 2015; Jimenez et al., 2021). However, these benefits are unevenly distributed and climate change may exacerbate existing inequalities, highlighting issues of 'blue justice' (Bennett et al., 2021). Socio-economic, geographic and cultural barriers can constrain both access to coastal spaces and the capacity to benefit from adaptation measures (Bennett et al., 2021; Cabana et al., 2023).

Moreover, wellbeing is not only an outcome but can also influence the future quality of blue spaces. Positive feedback occurs when benefits encourage continued use, stewardship and investment in amenities, reinforcing environmental quality and long-term health benefits. Conversely, negative feedback emerges when overuse, hazards or degradation reduce access and usability, limiting benefits. Climate adaptation can break negative loops and strengthen positive ones, protecting both environmental quality and the equitable delivery of wellbeing benefits. Climate adaptation is not a neutral or purely technical process. Adaptation decisions are shaped by political priorities, institutional power and uneven capacities for participation, meaning they can reproduce or even exacerbate existing inequalities (Adger et al., 2009; Eriksen et al., 2021). Decisions about which amenities are protected, enhanced or allowed to decline often reflect broader governance structures and power asymmetries as much as physical risk (Klein et al., 2005; Pelling and Garschagen, 2019). These dynamics shape everyday

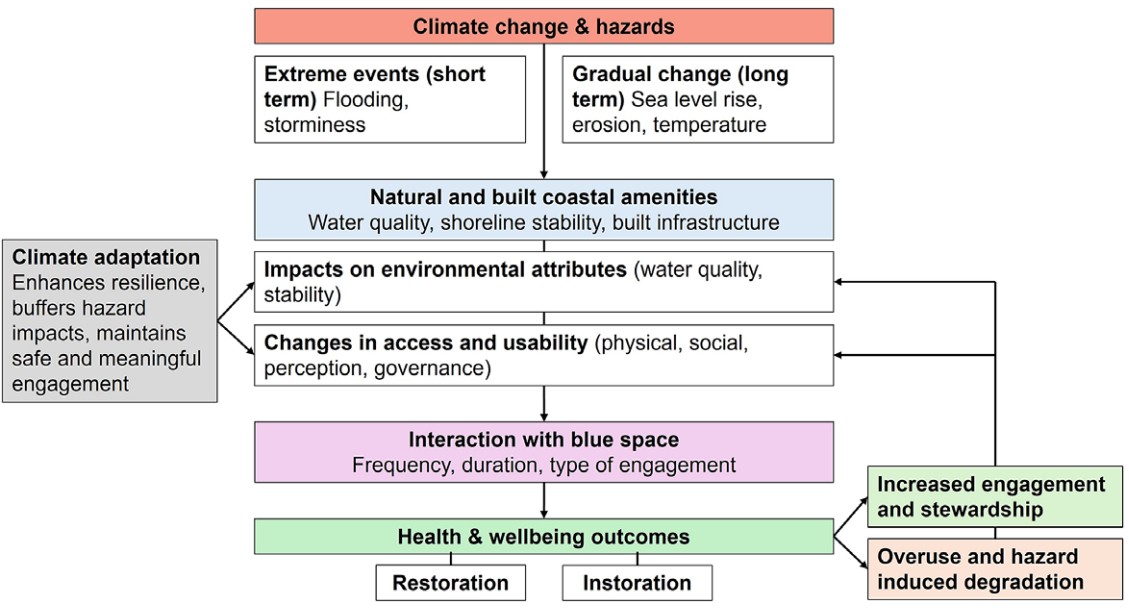

**Figure 1.** Conceptual framework linking climate change impacts on coastal amenities, environmental attributes, access and usability and human health and wellbeing.

patterns of coastal use, including distinctions between sustainable 'continued use' and 'overuse', where visitation exceeds ecological or infrastructural capacity. When overuse leads to degradation, the resulting loss of access and amenity disproportionately affects disadvantaged groups, who typically have fewer alternatives and less capacity to adapt (Szuster et al., 2023; Skiniti et al., 2024). Incorporating governance ensures that the framework not only accounts for environmental pathways but also for the social processes that shape who benefit from, or is excluded from, coastal amenities.

This framework is designed as a practical decision-support tool for coastal planners, policymakers and community organisations. It offers tailored entry points for each group's information needs. For planners, it highlights which physical processes shape the environmental conditions, access and usability of coastal amenities, helping them identify the evidence required, such as shoreline change projections, flood or erosion modelling or water-quality forecasts. Policymakers can use it to see how climate hazards may translate into reduced public access or the loss of key recreational assets, informing investment priorities and place-based adaptation strategies. Community organisations can draw on the framework to articulate local concerns, trace the pathways through which climate impacts affect wellbeing and support co-design processes.

Sections "Measuring coastal amenity and access"–"Climate adaptation" operationalise the framework by linking measurable indicators, hazard impacts and adaptation options to specific parts of the conceptual model. Together, these elements demonstrate how the framework can be used across scales, from site-level assessments by local managers to strategic planning by national agencies, to design climate-informed, equitable and wellbeing-focused coastal adaptation strategies.

## Measuring coastal amenity and access

The indicators outlined in this section operationalise the conceptual framework (Figure 1) by providing measurable ways to assess how climate-driven changes in environmental conditions, access and usability affect coastal amenity value. Summarising approaches to measuring coastal amenity and access are essential for understanding how human wellbeing is supported by coastal environments and the pathways through which exposure to blue spaces confers benefits. Assessment requires knowledge of both the physical characteristics of the environment and the capacity of different populations to use it (Natural, 2010). Key physical indicators include structural access (steps, ramps, slipways, boardwalks), transport and connectivity (distance to public transport, bike lanes, pavements), coastal morphology (gradient, width and sediment type influence ease of access; Hecock, 1970) and water quality (microbiological measures, short-term pollution alerts) (Anjum et al., 2021; Tiwari et al., 2021; Krupska et al., 2024). Records of re-routes following erosion or flooding provide insight into the vulnerability and resilience of coastal amenities. Safety is captured through lifeguard coverage, monitoring of hazards, signage and emergency equipment (Trimble and Houser, 2017), while affordability indicators, like parking fees, and inclusive facilities, such as changing facilities and toilets, shape equitable use (Kim and Nicholls, 2018; Job et al., 2023). Together, these metrics provide a framework for evaluating both the provision of coastal amenities and accessibility for diverse populations, even if patterns of actual usage or visitation patterns are unmonitored.

## Climate change and hazards

This section links major climate hazards, including storms and flooding, sea-level rise and erosion, water-quality degradation and heat stress, to the framework's pathways (environmental conditions, access and usability), and then to specific, practicable management options. The following are indicated for each hazard: (i) the amenity pathways at risk, (ii) thresholds that indicate a need to act and (iii) adaptation actions that decision makers can sequence within local or regional plans.

### Short-term impacts: Storms and flooding

Extreme events, such as storms and coastal flooding, disrupt the physical conditions, accessibility and safe use of coastal amenities,

generating immediate losses in recreational value and wider well-being impacts. These short-duration hazards damage steps, slip-ways, car parks and promenades, often forcing temporary closures or re-routing (Clark et al., 1998; Rizzetto, 2020). These also cause debris deposition, dune blowouts and short-term bathing water advisories from rainfall or combined sewer overflows, reducing amenity quality. Compound events, including heavy rainfall coinciding with high tides, intensify flooding (Lyddon et al., 2024) and extend recovery times. Such overlapping hazards generate cascading effects, for example, a washed-out car park-restricting access or impeding emergency response, and repeated storm events can create cumulative degradation at access points (Lawrence et al., 2020; Jenkins et al., 2023). Disruption to supporting infrastructure, such as public transport links or parking, further constrain safe and reliable coastal use.

In this framework, storms and flooding are rapid shocks that primarily disrupt access and usability, with secondary impacts on environmental quality. Storm-related impacts provide clear signals for when managers should shift from reactive to anticipatory adaptation. Repeated washout of access points and prolonged closures following storm clusters and compound flooding, now increasingly characteristic of UK and international coastlines (Zscheischler et al., 2018; Lyddon et al., 2025), indicate that existing maintenance approaches may no longer be sufficient. When closures extend beyond typical access expectations, secondary routes fail, or recurrent damage surpasses annual maintenance budgets, near-term actions are required. These include storm-response procedures such as temporary closures, re-routing, clear hazard or detour signage, alongside debris-management procedures, and temporary modular ramps or boardwalks to restore safe access quickly (Cucuzza et al., 2020). When such disruptions recur across seasons, anticipatory measures become necessary. These include elevating or relocating vulnerable access points, reinforcing or armouring promenades and car parks at critical sections and embedding compound-flood scenarios into emergency and maintenance plans (OECD, 2019). Within the conceptual framework, these triggers represent repeated failures in access and usability pathways, prompting redesign rather than repair to sustain wellbeing under more frequent storm disturbance.

### Long-term impacts: Sea-level rise and erosion

Rising sea levels and changing wave climates drive erosion, shoreline retreat, beach and dune narrowing, coastal squeeze and cliff instability (FitzGerald et al., 2008; Nicholls and Cazenave, 2010; Masselink et al., 2020) reducing usable beach width, complicating gradients and repeatedly burying or undermining access structures. Changes to offshore bars, depth of closure and local wave regimes further modify beach profiles, altering sediment distribution and coastal gradients (Durkin et al., 2025) further altering usability and scenic quality. In this framework, these are slow-onset pressures that progressively degrade environmental conditions and the physical nature of access, with cumulative losses in usability unless adaptation keeps pace.

Long-term morphological change signals when managers must shift from routine maintenance to proactive adaptation. Sustained beach-width loss, along with the recurrent burial or undermining of ramps and steps, or repeated closures from encroaching scarps or migrating dunes, indicates that existing designs are no longer keeping pace with shoreline change. When such thresholds are reached, managers can act at the local scale through path rollback or realignment, boardwalks over mobile dunes or adjustable ramps

that can be repositioned as profiles evolve, while managed relocation of small amenities is appropriate at persistent erosion hotspots (Cooley et al., 2022). At the shoreline scale, dune restoration, fencing and planting can enhance natural stability (Morris et al., 2018) and living shorelines or wetland restoration can maintain ecological and recreational value (Reguero et al., 2018; Leo et al., 2019). Targeted renourishment may be suitable in some areas despite ecological and financial trade-offs and need for ongoing investment (de Schipper et al., 2021), while hybrid nature-based and engineered defences may stabilise key access points during longer term transitions. These options should be selected using transparent criteria that weigh ecological impacts, nourishment cycles and the fair distribution of benefits and impacts across communities and access points, and tied to measurable morphological thresholds that trigger timely action. Within the conceptual framework, these thresholds identify where slow-onset change degrades environmental conditions and cascades into access and usability barriers; adaptation measures therefore target the pathways under pressure to preserve long-term wellbeing benefits.

### Water quality

Water quality mediates amenity value through microbial contamination, harmful algal blooms and other biological hazards that cause advisories or closures and can deter use via perceived risk (Heaney et al., 2014). Short-term pollution events (rainfall, CSOs, runoff) lead to advisories, while harmful algal blooms and jellyfish outbreaks affect comfort and safety (Berdalet et al. 2015; Igwaran et al. 2024; Karlson et al. 2021 Lim et al. 2023). Monitoring gaps (e.g., viruses, antimicrobial resistance organisms) mean some risks remain undetected under current indicator regimes, undermining both safety and confidence (Nappier et al., 2020; Clough et al., 2026). In the framework, water-quality shocks and chronic issues directly degrade environmental conditions, with knock-on constraints for access (closures) and usability (risk perception).

Water-quality degradation provides clear thresholds for shifting from reactive responses to structured adaptation. Exceedance frequencies, such as repeated bathing water advisories after moderate rainfall or recurring harmful algal blooms, indicate that current monitoring and infrastructure no longer protect public health or amenity value (Krupska et al., 2024). When such triggers occur, managers can prioritise source-control measures, including green–blue retrofits such as raingardens, and wetlands to intercept runoff, along with permeable surfacing and improved separation of foul and storm flows to reduce sewage-related contamination (Poggioli et al., 2024). Complementary forecasting and communication tools are essential: predictive models help anticipate short-term pollution risks (Clough et al., 2026), while real-time, alerts ensure equitable access to risk information (Stidson et al., 2012). Monitoring upgrades incorporating viral and antimicrobial-resistant indicators address persistent gaps in current bathing water assessments (Anjum et al., 2021; Environment Agency, 2026). Together, these measures strengthen perceived and actual safety, sustain public confidence and reduce the frequency and duration of climate-related water-quality disruptions. Within the conceptual framework, water-quality hazards primarily erode the environmental conditions pathway; these source-control, monitoring and communication actions re-establish safe conditions for access and use and prevent breakdown of wellbeing pathways.

### Heat stress

Rising temperatures not only increase the cooling value of coasts but also drive surges in visitation, overcrowding, heat-related safety risk. These dynamics shift peak-use windows and can exceed social or ecological carrying capacity where shade, water and supervision are limited, with inequitable impacts on users lacking alternatives. In the framework, heat primarily stresses usability (comfort, safety) and access (capacity limits).

Rising temperatures and intensifying heatwaves create clear thresholds for when enhanced management responses are needed. Temperature forecasts that exceed local comfort or safety limits, crowding that surpasses occupancy or emergency-response standards, and spikes in heat-related incidents indicate conditions in which coastal amenities may no longer operate safely or equitably (Perkins et al., 2012; Brown, 2020; Laino and Iglesias, 2023). When such triggers occur, amenity-scale interventions, shade structures, potable water, heat-risk and rip-current signage, temporary misting or first-aid capacity and surge lifeguard staffing, become essential (Kleerekoper et al., 2012; Kumar et al., 2024). At the same time, demand-management measures such as dynamic crowd management, real-time occupancy information and activation of secondary coastal sites help distribute visitor load, supported where needed by temporary transport adjustments (Coles, 2020; Zhou et al., 2021). Integrating coastal amenities into local heat-health plans, recognising their role as natural cooling infrastructure modulated by sea-breeze dynamics (Zhou et al., 2019; Yamamoto et al., 2024) ensures that staffing, opening hours and public alerts align with heat forecasts. Together, these measures maintain safe and comfortable use of coastal spaces while protecting social and ecological carrying capacities. Within the conceptual framework, heat stress first disrupts the usability pathway and then constrains access through crowding; these interventions directly stabilise those pathways to preserve coastal wellbeing benefits during extreme heat.

### Impacts on accessibility and equity

Ensuring equitable access to coastal amenities under climate change is a key challenge for adaptation because disruptions to access and usability disproportionately affect those with the fewest alternatives (Schüle et al., 2019; Phoenix et al., 2021; Fiorentino et al., 2024; Parsons et al., 2024). Less abled people face persistent obstacles such as soft sand, tidal steps, steep gradients (Job et al., 2023; Pool et al., 2023) and insufficient information about accessibility that constrain safe use even under present-day conditions. Climate driven changes further compromise coastal paths, slipways, stairways, car parks and ramps, reducing the availability of step-free or stable routes and narrowing the window of safe access.

Using the conceptual framework, these impacts indicate failures within the access and usability pathways: when repeated erosion blocks mobility routes, when gradients exceed acceptable safety thresholds or when storms significantly shorten access windows, managers can diagnose that adaptation actions are required. These thresholds help identify when low-cost measures, such as stabilised access mats or boardwalks are sufficient, and when more substantial interventions, such as realigning paths or relocating amenities away from erosion hotspots, become necessary. Assistive technologies can support mobility in some contexts, though persistent barriers related to storage, propulsion and realistic independence expectations mean they cannot substitute for structural improvements (Verdonck et al., 2024).

Climate change exacerbates existing inequalities, with disadvantaged populations, with limited financial or transport resources, often facing the greatest loss of access and fewest options for adaptation (Leichenko and Silva, 2014; Limaye, 2022; Thornhill et al., 2022). In contrast, wealthier groups may mitigate impacts by travelling further, using private facilities or relocating to more resilient areas. These dynamics raise critical questions about who loses access first and whose needs are prioritised in adaptation planning. Applying the framework, managers can identify where access loss intersects with social vulnerability and thus direct resources towards protecting the pathways most critical for maintaining wellbeing.

Adaptation decisions determine whose access is protected and whose is sacrificed, therefore governance practices play a decisive role in whether climate responses reproduce or reduce inequalities (Bennett et al., 2021; Parsons et al., 2024). Co-design with less abled users, low-income communities and local organisations ensures that decisions about realignment, erosion management or seasonal closures reflect the needs of those most dependent on public coastal access (Roukounis and Tsihrintzis, 2024). Integrating equity criteria into adaptation pathways helps ensure that interventions sustain the access and usability pathways for all groups, reinforcing the framework's emphasis on fair distribution of wellbeing benefits. Viewed through a justice lens, the framework helps practitioners identify where environmental degradation and social vulnerability converge, enabling adaptation choices that avoid reproducing power imbalances and instead foster equitable access and coastal wellbeing. By embedding blue-justice principles within the conceptual framework, managers can use it not only to diagnose where environmental pressures reduce access, but also to ensure that adaptation pathways protect the wellbeing of groups with the least capacity to absorb additional loss (Schlosberg, 2007; Bennett et al., 2021).

### Climate adaptation

This section examines how adaptation can reinforce or restore the environmental conditions, access and usability pathways that underpin coastal wellbeing. By linking specific hazards to intervention types and considering governance, equity and power dynamics, it shows how the framework can guide strategic planning, adaptive pathways and community-led decision making. Adaptation pathways can be used alongside the conceptual framework to address uncertainty and change over time. While the framework identifies how climate impacts disrupt key pathways underpinning coastal amenities, adaptation pathways focus on sequencing decisions and actions as critical thresholds are approached or crossed (Barnett et al., 2014). In this sense, adaptation pathways explicitly incorporate uncertainty, learning and flexibility, allowing future options to remain open rather than locking in responses early (Reeder and Ranger, 2011). Pathways emphasise inclusive and reflexive governance, supporting stakeholder engagement around when thresholds matters and how alternative actions can be triggered (Werners et al., 2021). Used together, the framework helps identify where and why interventions may be needed, while adaptation pathways clarify when and how interventions should evolve over time.

### Adaptation across scales

Climate adaptation in coastal areas offers a dual opportunity to mitigate hazards and sustain the ecological, recreational and cultural values of blue spaces, by leveraging feedback loops, adaptation

pathways and integrated nature-based solutions. Climate adaptation measures in coastal areas can break negative or enhance positive feedback loops (Figure 1) by protecting against climate-related hazards while preserving environmental quality, amenity value and continued access and usability (Griggs and Reguero, 2021). At the national scale, policy frameworks and planning tools, such as Shoreline Management Plans in the United Kingdom, guide long-term adaptation. Adaptive pathways provide economically efficient sequences of actions that respond to thresholds, needs and tipping points as they arise (de Ruig et al., 2019; Environment, 2021; Wentworth and O'Neill, 2021; Werners et al., 2021). A leading example is the Thames Estuary 2100 (TE2100) Plan, which applies pathways to manage flood risk over the coming century, sequencing interventions so that actions can be adjusted as sea-level rise and hazard thresholds are reached (Ranger et al., 2013). Globally, similar approaches underpin coastal planning in the Netherlands' Delta Programme, New Zealand's coastal hazards guidance and U.S. resilience strategies under NOAA. While some systems operate on stronger statutory foundations, they share a common emphasis on threshold-based sequences of interventions under uncertainty. These approaches set the context for implementing both nature-based solutions, such as dune restoration, wetland creation and living shorelines, which deliver dual benefits of natural protection and recreational value (Reguero et al., 2018; Leo et al., 2019). Increasingly, such strategies reflect a wider shift from reliance on grey infrastructure towards green or hybrid approaches, which not only provide protection but also enhance ecological function and maintain the recreational, cultural and aesthetic value of coastal amenities (Reguero et al., 2018; Leo et al., 2019; Schoonees et al., 2019).

While national-scale tools such as SMPs and resilience strategies provide high-level strategic direction, many of the adaptation mechanisms relevant to coastal amenities operate at much finer spatial scales. Site-specific interventions, such as path realignment, modular ramps or accessible design modifications, address the day-to-day usability and safety of individual beaches, promenades and access points. At regional and community scales, participatory, community-led initiatives ensure that adaptation balances environmental, social and economic priorities, fostering local stewardship and responsiveness to diverse user needs (Jones and Russo, 2024). Broad participation and diverse tools, including place-based digital technologies, create new ways for communities to access and engage with the sea (McKinley et al., 2021; Willis and Gupta, 2025). Finally, at the scale of everyday amenities, effective adaptation depends on practical interventions that safeguard usability such as adaptive path alignments, modular or movable ramps and boardwalks, inclusive accessibility features, dynamic signage, digital alert systems for water quality, heat and rip currents and shade or cooling infrastructure. Designing for resilience across these scales, from national policy to local amenities, ensures that coastal spaces remain accessible, usable and valuable for wellbeing under climate change. The framework enables practitioners to link these local actions to broader strategic pathways by identifying how changes at the amenity scale fit within longer term trends in shoreline dynamics, flooding or heat risk.

### Practical constraints and design considerations

Building on these considerations, the management of coastal amenities requires a careful balance between access, safety and environmental protection (Thompson, 2007). Expanding access is not always feasible, as coastal typologies such as rocky shores, macrotidal

estuaries and cliffs pose inherent risks (Kennedy et al., 2013), while sensitive habitats, including bird nesting areas and wildlife breeding grounds, require protection. Strategic planning therefore needs to focus on maximising safe and equitable accessibility where it is most valuable, particularly in urban-adjacent blue spaces that function as key neighbourhood assets (Hayes and Lyddon, 2025). In this way, coastal spaces can continue to support public use and wellbeing, while ensuring that conservation and safety goals are not compromised.

### Governance, trade-offs and power dynamics

While climate adaptation is often framed as a universally beneficial response, it is also a politically contested process that can generate uneven outcomes. Adaptation choices determine who experiences risk reduction, which amenities are preserved and where resources are directed; decisions that are shaped by institutional authority, competing values and unequal access to planning processes (O'Brien, 2012). Adaptation can create winners and losers, with investments frequently favouring high-value tourist or economically productive areas while under-resourcing sites that hold everyday social or cultural significance for local communities (Juhola et al., 2016). These trade-offs highlight the need for adaptation pathways and thresholds to be co-developed through inclusive governance that not only reflects community priorities, but also technical or economic criteria (Wise et al., 2014). Integrating these power-sensitive perspectives into the framework ensures that it can be used not only to assess environmental change but also to evaluate how adaptation strategies may reinforce or redress inequalities in coastal wellbeing. Preserving coastal wellbeing requires planning that integrates climatic, ecological, social and governance dimensions. The conceptual framework identifies where climate pressures affect environmental conditions, access and usability at specific amenities, while adaptation pathways provide a complementary structure for sequencing interventions over time and across scales. Used together, the framework pinpoints emerging pressures and the pathways approach clarifies when and how actions should be triggered as thresholds are reached. Embedding coastal amenities within pathways planning helps prioritise measures that protect not only safety but also the restorative, cultural and social value of blue spaces. This combined approach highlights where early, low-regret actions, such as enhancing accessibility, improving cooling functions or safeguarding key amenities, can maintain long-term wellbeing benefits. By linking conceptual insight with strategic sequencing, it offers a clear foundation for coastal planning that places human wellbeing at the centre.

### Research gaps and future directions

Despite growing recognition of the importance of coastal amenities, significant research gaps remain in understanding how climate change will reshape their value, accessibility and health benefits. Longitudinal studies are particularly scarce, limiting our understanding of how access to blue spaces under climate stressors (flooding, erosion, extreme heat) affects wellbeing over time. Causal evidence linking specific adaptation interventions, such as the combined provision of shade, lifeguards and water points, to health outcomes on hot days is still limited, and long-term social monitoring of adaptive measures, including coastal path realignments and roll-back, remains underdeveloped. There is also a pressing need for research on equity and disability-inclusive design, including trials of accessible coastal infrastructure, to ensure that

adaptation strategies support diverse user groups and uphold principles of 'blue justice'. Integrating social, ecological and engineering perspectives through coupled physical–ecological–social models allows the evaluation of amenity under scenarios of sea-level rise, erosion and heat, providing a more comprehensive understanding of potential futures. Combined with co-design processes and community engagement, these approaches can ensure adaptation measures are both socially responsive and ecologically sound. Addressing these gaps will be critical for guiding fair, evidence-based and holistic coastal adaptation policies that protect both physical and social value in a changing climate.

## Conclusion

Climate change threatens the environmental conditions, access and usability of coastal amenities, limiting safe engagement with blue spaces and thereby undermining wellbeing, cultural identity and community resilience. The conceptual framework demonstrates that climate change affects wellbeing indirectly, through its impacts on environmental quality and on people's ability to access and use coastal blue spaces. Adaptation, while not essential for every form of benefit, is critical to safeguard ecological integrity, preserve usability and enhance the magnitude and reliability of wellbeing outcomes, particularly for communities with fewest alternatives.

By integrating climate science, local knowledge and inclusive planning, this framework provides a structured, transferable basis for understanding and managing climate impacts on coastal amenities as essential public health infrastructure. It is the foundation for just, place-based adaptation and highlights where future research and practice must work across disciplines, governance and communities to sustain the cultural, social and health value of coastal environments for current and future generations.

**Open peer review.** To view the open peer review materials for this article, please visit http://doi.org/10.1017/cft.2026.10032.

**Data availability statement.** This research did not involve the collection or use of any empirical data. There are no datasets associated with this manuscript.

**Author contribution.** The author solely conceived the study, developed the conceptual framework, conducted the literature review and analysis, created all figures and tables, wrote and revised the manuscript for submission.

**Financial support.** This research received no specific grant from any funding agency, commercial or not-for-profit sectors.

**Competing interests.** The author declared no potential conflicts of interest with respect to the research, authorship and/or publication of this article.

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
