## [Reviewer Report]

This manuscript provides insight into a relevant topic that is certainly within scope of the journal by presenting a framework to enable a more comprehensive understanding of how climate change impacts coastal amenities, and by extension, human health and wellbeing. The paper is very UK-centred.

The impact statement clearly states that this work is relevant to policy makers, planners, and community organisations but the manuscript does not lay out entry points for how these stakeholders can or should use the framework. How this work feeds into the needs of these stakeholder groups is not revisited in the body of the manuscript.

The introduction (section 1) provides context for blue spaces as well as a good argument to link coastal amenities with human wellbeing. However, the presentation of the framework (section 1.1) doesn’t provide a clear objective of how it should be used or the contexts under which this research has been undertaken. Overall, the framework aims to establish/describe a landscape where coastal amenities as a proxy for wellbeing is considered as a component of a coastal management regime responding to both gradual change and extreme events under climate change. The framework, presented as a diagram, is not expanded on in the text providing little reference as to its purpose/ intended outcome or who could usefully engage with it as a management mechanism. Providing a clear use pathway for this tool would improve the narrative of the manuscript.

Blue justice is raised in the context of exacerbation of existing inequalities and access to amenities. The manuscript also appears to imply that climate adaptation is the solution to these inequalities. Adaptation has repeatedly been described as series of highly contested processes steeped in issues of inequality and exploitation (e.g. Pelling & Garschagen, 2019 and many others). The manuscript doesn’t critically engage with the doing of adaptation (even in section 4) or how potential power dynamics may impact or distort the framework as it relates to different management scenarios. Whilst the dynamics of positive and negative feedback within the system is referenced in the text (page 3, line 54-59), there is no attempt to qualify differences between ‘continued use’ and ‘overuse’ for example. The manuscript would be enhanced with a critical analysis of the elements of the framework, providing much needed clarity and context for the intended user of this tool.

The subsequent sections of the paper do not directly engage with the elements of framework, creating a disconnect between the diagram and the rest of the manuscript.

Section 2 provides a range of indicators for amenities at a site-specific scale but does not connect the indicators to the framework – if that is the intention.

Section 3, with the exception of section 3.5, focusses on climate hazards and the impact of these hazards on costal amenities and infrastructure. This text provides clear descriptions of the problems that climate hazards are either creating or exacerbating but the text lacks insightful elements of the ‘so what’ question. In some places the text picks up on key words in the framework but the manuscript would benefit from more explicit and detailed reasoning linking the what to a potential suite of adaptation actions that may provide management options under climate change impacts. Section 3.5 (pg7, line 7-10) raises important questions around inequalities in adaptation planning. The manuscript would be hugely improved if it critically engaged with these questions rather than just stating them. This would ensure that the manuscript adds to existing literature rather than just presenting possible research avenues.

It is unclear if the manuscript is advocating for adaptation pathways (pg7, line 21) to be used in conjunction with the framework provided or instead of it. It is also unclear as to the scale at which the adaptation mechanisms discussed in section 4 are relevant to the very site-specific coastal amenities described. The final paragraph in this section (pg7, lines 52 onwards) appear to be at the heart of what this manuscript wishes to present – and by far its most relevant argument. It is unfortunate that the points made here are not expanded on more critically and not centred within the rest of the text. Reworking the manuscript and providing direct and distinct insight into how to go about strategic planning for coastal amenities, and by extension human wellbeing, would result in a much more impactful paper.

---

## [Reviewer Report]

This article is an important contribution that would benefit from some minor changes.

Page 6 lines 23-27: I don’t see the connection here between urban heat island effects and sea breeze conditions.

Page 8, line 37: “The conceptual framework shows that climate change impacts wellbeing not directly…” should be reworded to explicitly re-state the bounds of this assertion (environmental quality and accessibility of coastal blue spaces) in the first part of the sentence

Proofread for typos, including double spaces between some sentences and not others. A few grammatical fixes/incomplete sentences need to be corrected throughout:

Page 1, line 51: “This framework offering...”

Page 5, line 4: “These events can also debris...”

Page 6, line 59: “Climate change exacerbates these disparities, with disadvantaged populations, or those with…”

---

## [Editor Report]

Dear Charlotte

Firstly, I apologies for the time it has taken us to secure review of your manuscript.

As you can see, both reviewers consider this to be an important topic and relevant to the journal. I would like to echo my own wish to see the paper published. One reviewer has come back with significant comments and whilst I agree with the thrust of the points made I believe that that the concerns are addressable and that the paper can be improved to a level whereby I will be able to recommend acceptance. The reviewer provides feedback that clearly indicates where they consider the text to have weaknesses (while also recognising the relevance and significance of the topic and research) and clear direction for how these areas should be amended and improved to meet publication standards.

As a UK-focussed piece of work, I think it is important to place how a UK situation and circumstance with the outceoms you have found have an international context.

I very much hope that you will consider re-submitting the manuscript.

---

## [Reviewer Report]

The author must be commended for the edits to the manuscript and willingness to engage with the complexities of well-being driven adaptation as a whole. There is a much clearer sense of how the framework can be applied, by whom, and at what scales it could be used. There is also a greater degree of clarity around specific adaptation actions that could be used under a range of climate hazards and potential cascading hazards and how the framework can provide an structured basis for decisions. The more comprehensive framing of blue justice and examination of equity and power dynamics of adaptation actions across the elements of the framework provide increased positionality towards disproportionately impacted groups and detailing the need to centre the justice lens in adaptation efforts.

I would suggest that the introduction of adaptation pathways as a complementary method (section 4, line 60) could be more comprehensive to allow readers to distinguish between the two. The inclusion of references such as Reeder and Ranger (2011) and Werners et al. (2018) would help this. Whilst the role of adaptation pathways is discussed in parts during the remainder of the manuscript, an explicit description would be beneficial especially considering the call for the introduction of inclusive governance within the delivery. This would also provide important entry points into the literature for readers not as aware of adaptation pathways, clarifying the way in which pathways and the developed framework can work together.

As climate change continues to pressure coastal governance systems into complex, and often potentially conflicting decisions, frameworks such as the one presented in this manuscript can provide important touchpoints to ensure justice and equity are not lost in the discourse.

---

## [Editor Report]

Dear Charlotte

I want to thank you for making the effort to revise the manuscript in the manner that you have - as clearly noted by the reviewer (who had recommended to reject) - and addressing so comprehensively all the reviewer comments. I believe this is now a very strong and important paper and I am very happy that it will be published in Coastal Futures. I have given a ‘Minor revision’ recommendation because as you will note the reviewer has made one further small suggestion that I think merits your consideration - I think it would require very minor alterations to the text and the inclusion of a couple more references (that are provided). Whether and how to address this comment I leave to your discretion and upon return I will be marking the manuscript for acceptance.

Kind regards, Martin

---

## [Editor Report]

The author is thanked for addressing all the reviewer comments and I am happy to recommend that the manuscript is accepted.